# Analysis of Bilateral Air Services Agreement Liberalization in Australia

Iryna Heiets *[ID], Richard C.K. Yeun [ID], Wim J.C. Verhagen [ID] and Jiezhuoma La [ID]

School of Engineering, RMIT University, Building 57, Level 3, 115 Queensberry St., Carlton, VIC 3053, Australia; richard.yeun@rmit.edu.au (R.C.K.Y.); wim.verhagen@rmit.edu.au (W.J.C.V.); s3633823@student.rmit.edu.au (J.L.)
* Correspondence: iryna.heiets@rmit.edu.au

**Abstract:** This paper examines an assessment of the level of air transport services liberalization in Australia in order to generate recommendations on what key market access features of Air Services Agreements should be revised to reflect the changes in air transport characteristics, including the increase in air cargo traffic during the COVID-19 period. The different variants of the key market access features of ASA, levels of air transport liberalization and the extent of air transport service liberalization between Australia and 104 partner countries were analysed using descriptive study, comparison analysis and the ALI index. The ALI index is calculated for four different weighting schemes. Passenger capacity in 41 bilateral agreements contain restrictions of frequency, capacity and aircraft type. The analysis of cooperative arrangements indicated that Australia has a single aviation market only with New Zealand. The cargo capacity analysis identified different types of capacity restrictions based on weekly cargo service, volume, destinations, designated airline and aircraft types. In conclusion, cargo capacity analysis illustrates that the level of liberalization is high, but the air services agreements between Australia and other countries in the first and second cargo capacity groups should be revised to reflect the increase in air cargo traffic during COVID-19.

**Keywords:** air services; bilateral agreements; liberalization; Air Liberalization Index; comparison analysis; COVID-19

## 1. Introduction

International airline operations are governed by a complex web of Air Service Agreements (ASAs) or Bilateral Air Service Agreements (BASAs) developed under the principles of the Chicago Convention, 1944. The International Air Transport Association (IATA) estimates over 3000 such agreements are in place today and only 200 of these agreements are responsible for approximately 75% of the total traffic in the world [1]. A BASA prescribes the terms agreed between two countries and defines the degree of market access a carrier will have in the particular market. They are different for each country with the most commonly agreed elements being traffic rights, authorized points, capacity, pricing, designation and other clauses relating to operative agreements [2].

International airline operations were affected heavily over the last years through the effects of the COVID-19 pandemic. This pandemic has had a negative impact on people's lives, including increased unemployment rate [3]. Many countries and governments had to react swiftly and lockdown their country's border or a specific region after the breakout of COVID-19. In the face of strict regulations and restrictions that governments carried out for COVID safety, leading companies in the aviation industry in various countries lost a lot [4]. According to ICAO [5], compared to 2019, the industry offered around 50% fewer seats, carried about 2.7 billion fewer passengers, and airlines lost approximately USD 370 billion of gross passenger operating revenues in 2020. This effect has been pronounced in many regions and markets, including countries, such as Australia, where commercial aviation activities were significantly impacted due to restrictions in air transport [6], with

international revenue expected to decrease by 31.5% over 2020 and 2021 [7]. One of the notable phenomena observed in air transport during this period has been the shift towards air cargo traffic [8], which has expanded to meet surging demand for a variety of goods, including vaccine transport.

Given the swift contraction of commercial air passenger transport and the expansion of air cargo traffic, there is a need to assess and, where required, revisit BASAs to ensure that the right set of agreements are in place to allow operators to respond to rapidly changing market conditions. From the industry perspective, there is a clear desire to have BASAs in place that serve the ongoing recovery and future expansion of commercial air transport; in other words, a means to assess and action liberalization of air transport services is required.

From an academic point of view, several frameworks and indices are available to support the assessment of air transport liberalization. Nevertheless, there are two distinct application-oriented gaps in the state of the art which can be identified:

1.  There has been no comprehensive and systematic effort to characterise market liberalization in the context of recent developments, prompted by the COVID-19 pandemic;
2.  Academic studies that apply and validate liberalization indices towards specific air transport markets tend to focus on economic effects, and, furthermore, lack application in various markets.

As such, this research aims to provide an assessment of the level of air transport services liberalization specific to Australia. Furthermore, it aims to generate recommendations about what key market access features of Air Services Agreements should be revised to reflect the changes in air transport characteristics, including the increase in air cargo traffic, during the COVID-19 period. To guide this research, several research questions are proposed:

(1)  What is the current level of air transport services liberalization in Australia?
(2)  To which extent and in which direction(s) should existing ASAs be revised to respond to ongoing market developments?

The structure of the paper reflects this aim and the associated research questions by first setting out the theoretical context in Section 2, including the substantiation of the identified research gaps. Subsequently, the methodology underlying this research is presented, which comprises a mix of comparison analysis, descriptive study and Air Liberalization Index (ALI index) application (Section 3). The research utilizes secondary data derived from different aviation and government agencies, such as the Department of Infrastructure, Transport, Regional Development and Communications. Subsequently, an analysis of bilateral air services agreements in Australia is performed in Section 4. Finally, conclusions are drawn and recommendations for future research are given (Section 5).

## 2. Literature Review

### 2.1. Air Transport Liberalization

Air transport is one of the most regulated industries globally, with much of the regulation being risk mitigation. The economic regulation also enables some of the most prolific airlines to operate routes at prices determined by the nationality owning the airlines. With the American Deregulation Act and the revoking of the European Union's BASA, it enabled the air transport market to open up internationally [9]. They studied air transport liberalisation and its impacts on airline competition and air passenger traffic. With open competition comes opportunities to new routes and tourist destinations. The result is a more acceptable re-definition of the air transport sector. From a global perspective, Piermartini and Rousová [10] used a sample of 184 countries and identified that liberalisation has brought up to a 10% increase in global passenger traffic. The paper also led to similar findings led by Fu, Oum and Zhang [11]. The authors analysed air transport liberalisation and its impacts on air passenger traffic and airline competition. The study evaluated the impacts of liberalisation policies on different factors, such as economic progression, air traffic capacity, and traffic flow patterns. It is concluded that the effect of liberalisation

led to substantial economic and traffic growth, attributed to the increased competition and efficiency gains within the industry. Moreover, liberalization has allowed airlines to optimize their networks inside and across different markets, resulting in changes in traffic flow patterns. The introduction of the Open Skies Agreements (OSA) in the United States had increased the shares of imports arriving by air from countries that signed the agreements [12]. Another paper measures the effects of the OSA on service export and import trades using an econometric model for Canada. The findings showed that with OSAs, there was an increase in Canada's services export and import and increasing commercial services trades with the partner countries [13].

The literature review aims to show the liberalization of the air transport market mainly through the bilateral agreements and the resultant outcomes. Bilateral agreements between countries always cited the business's intention to provide the right combination of rules and leeway for the airlines. Notwithstanding this, the evolution of bilateral agreements saw many—especially first world—countries open up the market for international business purposes [14]. Additionally, liberalization also changed people's travel habits with increased demand to tourist destinations as they opened up and airlines profiting with massive new business opportunities. Conversely, Finger and Button [15] suggest that with deregulation, the contracted airline market plus government control made the business unprofitable.

Air transport in most economies is a highly monopolistic business [16]. The efficiency of the air transportation system, its demand especially from the international market, encouraged more bilateral agreements worldwide and further deregulation to enable the industry to grow [17].

The quality of service is measured using price, quantity, the return of value to the customers and the market it operates [17]. The economic benefit of the airline industry became important as a result. In the Asian market, air transport regulations remain very prescriptive with restrictions on airline ownership and routes. However, changes are being made through plurilateral agreements. This has opened up the economies and convinced the governments of the importance of deregulation. Moreover, the economic implications with the opening of new routes enabled the Asian and Pacific regions to trading access with international markets [18].

Adler, Fu, Oum and Yu [19] presented the differentiated Bertrand airline network and high-speed rail game, including the effects of international air transport liberalization. Their presented model illustrated the benefits of the air transport liberalization for both customers and airlines. According to Goetz and Graham [20], the idea that globalization and liberalization have resulted in excessive air traffic growth and impacted directly on sustainability issues.

Although the price of an air ticket remains high, in part due to the included charges from airports, passenger demand for air travel especially to tourist destinations continues to surge. The deregulation of the airline industry helped spur economic growth for many countries as they benefit from bilateral agreements and extension agreements. Markets in China, Australia and Africa developed and benefitted by extension of the agreements as a result [21] Deregulation also increased a demand for passenger air travel. This sparked further economic growth in different areas of the industry.

The overall impact of air transport liberalization is an opening of new markets and the creation of a booming new aviation industry. Notwithstanding this, there are many other elements to manage and consider alongside this growth, such as the financial and safety aspects. For example, the high costs of purchase or lease and operation of an aircraft. After COVID-19, passengers also will expect a safe, high quality and professional level of service and these expectations invariably may dampen the speed of growth and development of the industry. However, travel bubbles are presented as a possible future for aviation [22]. For instance, Japan has announced plans to create a travel bubble with Asian countries, such as Cambodia, Laos, Malaysia, Myanmar, and Taiwan, in September 2020 [23]. Although such plans have been overtaken by the re-opening of the market as per 2022, the point remains that BASAs should have sufficient flexibility to allow for such measures in the event of

future major market disruptions. As an example of an area requiring such flexibility, cargo transportation can possibly be seen as a new opportunity for the recovering aviation industry [8,24].

### 2.2. Australian Aviation Market and COVID-19

Since the 1980s, in the context of global liberalization, airports have been redefined as commercial enterprises [25] that had given the opportunities for low-cost-carriers to enter the Australian domestic air travel market [26]. It is no doubt that the commercial aviation plays an extremely important role in the tourism and goods import and export of Australia, which has a direct connection to Australia's economic development. After realizing the great role of commercial aviation, the Australian government has been vigorously promoting the development of commercial aviation in recent years. In 2014, the Victorian Government declared about a AUD 500,000 grant plan for encouraging commercial innovation within the aviation industry [27]. Not only is there strong domestic promotion, but the development of Australian commercial aviation is also inseparable from external influences. The relationship between the COVID-19 pandemic and aviation is interactional. On the one hand, with the outbreak of the COVID-19 pandemic, plenty of countries chose to close their borders or imposed severe travel regulations [28]. For instance, Australia restricted several kinds of people entering the border, including international students. Besides, the pandemic resulted in many airlines relying on governmental assistance to avoid bankruptcy [29]. In Europe, in order to keep slots at airports, airlines have been forced to operate on regular routes with quite a low load factor and revenue [30]. What is more, on the economic scale, COVID-19 has caused quite a large amount of loss in aviation revenue. On the other hand, the influence of aviation on the COVID-19 pandemic is also equally massive. Aviation and other transport systems also sped up the spread of this pandemic [31]. Air transport also increases person-to-person transmission risks [32].

During the COVID-19 pandemic, the Australian government implemented strict border restrictions, which greatly reduced international aviation activities. Even travel across states became very difficult. Hundreds of airliners were forced to park at the airport due to travel restrictions [7]. The COVID-19 pandemic made it difficult for airlines in Australia to survive. Some airlines had to shut down their services in face of huge financial pressure. In September 2020, Virgin Australia Airlines, which was the second largest airline in Australia, entered bankruptcy protection [33]. Qantas, the largest airline in Australia, also found it hard to tide over. The Australian government provided AUD 165 million to Qantas and Virgin Australia in April 2020 to maintain the operation of major domestic routes and other essential services [6]. Reports claims that the federal government also provided additional assistance to the commercial airlines in the aviation industry in order to maintain the regional aviation activities as much as possible (2020), which made the total assistance to the aviation industry up to a staggering AUD 1 billion in 2020 [32]. The COVID-19 pandemic has also brought great inconvenience to citizens' daily life. In addition to strict state border restrictions, the epidemic also threatened people's lives when they travel. To this end, the Australian civil aviation industry has issued the Domestic Passenger Journey Protocol (2020) to ensure passenger safety during the pandemic, which provided a clear guidance for reducing the risk of the pandemic spreading and dealing with the principles of domestic aviation activities.

### 2.3. Gaps in the State of the Art

From a methodological point of view, several methods and indices are available to characterize air transport (market) liberalization, as mentioned in more detail in the next section. However, the current state of the art lacks two key application-oriented considerations:

1. There has been no comprehensive and systematic effort to characterise market liberalization in the context of recent developments prompted by the COVID-19 pandemic. As highlighted above in Sections 2.1 and 2.2, various major developments prompt a re-

view of liberalization characteristics to enable increased flexibility and responsiveness to events with a global reach, such as COVID-19;

2.  Academic studies that apply and validate liberalization indices towards specific air transport markets tend to focus on economic effects, and, furthermore, lack application in various markets. In particular, the Australia/Oceania market has not been considered in detail in prior work.

The current research aims to address these gaps in the state of the art and thereby deliver a novel, application-oriented contribution.

## 3. Methodology

The impact of liberalization on air services can be estimated and analysed by using different methods and approaches. The two commonly used are the sham approach, which is proposed by Micco and Serebrisky [12] to use fake variables to demonstrate the effects of liberalization of air cargo markets on transport costs, and the other is the Ordering and Score framework. Under the primary quantitative methodology, Finger and Button [15] produced a fake variable to demonstrate a liberal or non-liberal system of global air transport strategy, dissecting the effect of global air transport strategy on cost. Micco and Serebrisky [12] utilized a sham to demonstrate ASAs, which takes the worth for Open Skies Agreements (OSAs), and for non-OSAs, to gauge the effect of advancement of the air freight market in the US on transport costs [15]. This methodology could recognize the vehicle costs influenced by OSAs, while the effect of non-OSAs with different levels of transparency could not be estimated. It is, as yet, hard to quantify the advancement level of a country's global air transportation strategy as reflected by reciprocal air administration arrangements (ASAs) [15].

In January 2013, the Secretariat of the World Trade Organization released an analytical tool that enables users to obtain information on an economy's network of bilateral air services agreements and associated passenger traffic flows. The WTO has selected the main market access features and construction of an Air Liberalisation Index, which significantly impact market access, i.e., designation, withholding, tariffs, capacity, traffic rights, absence of exchange of statistics, allowance of cooperative arrangements (Figure 1) [34].

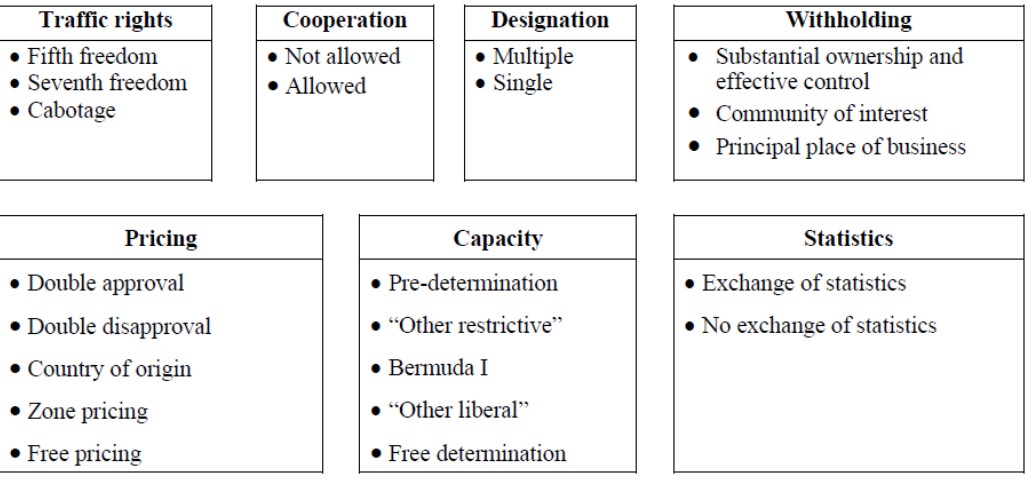

**Figure 1.** Elements of ASA agreement.

The Air Liberalization Index (ALI), built by the WTO Secretariat [35], is an expert-based list whereby the loads doled out to the various arrangements of air arrangements are characterized in meeting with a gathering of specialists in the flying industry with the view to catch the family member significance of each arrangement in changing the area [15]. The ALI ranges somewhere in the range of 0 and 50, where 0 is related with the most prohibitive arrangement and 50 signifies the most liberal understanding, as shown in Appendix A Table A1 [36].

The ALI can be expressed using the following equation:

$$ALI = \sum_{i=1}^{7} ALI\_element\_score_i \qquad (1)$$

With the 7 element scores representing individual scores for the 7 elements highlighted in Figure 1, as expressed across four weightings. With respect to the latter, four diverse weighting plans were proposed, in this way beginning with four distinctive files. The weighting plan of the supposed standard ALI index allocates a load between 0 and 8 to every one of the 7 parts of ASAs. Every one of the three other files underlines one explicit element of ASAs, specifically the conceding of fifth opportunity traffic rights retaining and assignment condition. Specifically, the ali_5thfreedom relegates a load of 12 to the fifth opportunity. The ali_ownership allots a load of 14 to the arrangement that permits unfamiliar aircrafts to support a country in the event that their chief business environment or generous proprietorship and compelling control is in the unfamiliar country [15].

The adopted methodology for this research comprises a mix of comparison analysis, descriptive study and Air Liberalization Index (ALI index) application. The ALI is selected because of its broader application in multiple studies and contexts; however, to the best of the authors' knowledge, it has not been applied in academic studies relative to the target focus of this research, i.e., the Australian market. Although the ALI gives a systematic and traceable way to quantify liberalization of a market, in and of itself it is not sufficient to fully characterize a selected market. As such, the research methodology aims to address the posed research questions by adding descriptive analysis and comparison analysis to fully ascertain both the liberalization of the Australian air transport market, and to identify recommendations for future changes.

## 4. Results

As the research aims to investigate the air transport services liberalization and relevant impacts on the air transport market in Australia, the study has been conducted from four aspects. Based on the definition of air bilateral agreements, all the air bilateral agreements between Australia and other countries have been analysed in Section 4.1 to identify the similarities and differences. The key findings of the analysis include the current state and key features of the Australian air bilateral agreements. Then in Section 4.2, we analysed the different agreements of air cargo services between Australia and other countries. Three special cases have also been analysed specifically to support the investigation of the key features of the Australian air bilateral agreements. At last, based on the Air Liberalization Index (ALI), the degrees of air liberalization between Australia and other countries have been calculated to identify the levels of Australian air liberalization. The findings and discussions in this chapter can be effectively used to identify the current status of Australian air transport services liberalization and it also enables the Australian aviation industry to identify how to mitigate the negative impact of COVID-19 in the post-pandemic period and help the industry perform as before.

### 4.1. Analysis of Bilateral Air Services Agreements in Australia

This section presents the analysis of all air bilateral agreements between Australia and other countries, as well as identifying the levels of air service liberalization. All information used in the section was sourced from the Department of Infrastructure, Transport, Regional Development and Communications of Australia [37]. By analysing all Australian air bilateral agreements provided by this department, the seven ALI elements of each agreement have been obtained and the specific air rights between Australia and other countries can be identified. The findings enable the calculation of the level of air liberalization of each air service agreement, and the guidelines for future development of Australian air transport can be generated based on the results of this study. Over the last 30 years, many countries globally have undertaken a process of liberalization of their air transport industry. Up until

around 1990, Australia has been imposing quite detailed and prescriptive regulations on international airlines flying to Australia. Currently, Australia has air services agreements with more than 100 countries. Over time, fare regulation has been slightly relaxed. The last 10 years saw occasional reviews of international aviation policy, but however, the aviation market in Australia remains tightly regulated by bilateral agreements. These agreements control carrier and route designations, seat capacity of aircraft used, prices, level of cooperation and tariffs.

Based on the analysis on the grant rights, 55 agreements between Australia and other countries mentioned that each contracting party grants to the other contracting party the rights for international air services, such as to fly without landing across its territory, to land in its territory for non-traffic purposes and to land in its territory for the purpose of taking on board and discharging international traffic in passengers, cargo and mail while operating an agreed service. Thus, airlines operate based on four freedoms of the air. However, each agreement has additional information that shall also enjoy the fifth right specified in the annex. The bilateral agreement between the Government of Australia and the Government of Canada relating to Air Services presents that the route to be operated in both directions by the designated airline of Canada and Australia can have Intermediate Points only in San Francisco, Honolulu, Tahiti and Fiji. Moreover, it is mentioned that any point or points specified above may be omitted on any or all services, but all services shall originate or terminate in Canada or Australia and points to be named by either contracting party may be changed on six months' notice given to the other contracting party. According to the bilateral agreement between the Government of Australia and the people's republic of China relating to Air Services, Australian airlines can operate between any point in China and vice versa.

The designated airline of Fiji can only operate in both directions between Brisbane, Sydney and Melbourne. Based on the agreement with Malaysia, the points of departure can be in the in the Commonwealth of Australia (Papua New Guinea, Christmas Island, and Cocos (Keeling) Islands), intermediate points are located in Indonesia or Singapore and the point in the territory of Malaysia is only Kuala Lumpur. Any cities in Malaysia can be points of departure for Malaysian airlines and the airports of arrivals should be located in Darwin, Perth, Melbourne and Sydney.

The designated Australian or Malaysian airlines may call at one or more points not indicated on the routes but shall not have the right to uplift or discharge at any such point of points traffic. Air service operations between Australia and the Philippines are based on the fourth air freedom but designed airlines from both countries can operate without any restrictions on departures, intermediate, arrives and beyond points.

As the most special case, the air agreement between Australia and New Zealand has obvious differences from others. One of the differences is about SAM airlines. The Australian airline can fly in both of the two parties' domestic parties, fly between any point and they need not be a designated airline, which is unique in all Australia air service liberalization agreements. Moreover, airlines from both countries can keep (or change) their aircraft or flight number and advertise during the service and can fly from the third parties via the origin country and any intermediate point(s) to the other party and then beyond the other party. According to the analysis of designation and authorisation of airlines rights, 102 agreements include the information that each contracting party shall have the right to designate as many airlines as it wishes to operate the agreed services and to withdraw or alter such designations (Figure 2).

Thus, cooperative arrangements are mostly allowed. A designated airline may be either an operating airline or a marketing (non-operating) airline, or both. As an exception, the Australian government shall have the right to designate in writing to the Socialist Republic of the Union of Burma only one airline for the purpose of operating the agreed services on the specified routes. Air traffic between Australia and Ethiopia can only be operated by two designated airlines. Therefore, we can conclude the Australian borders open for multiply airlines from more than 100 countries.

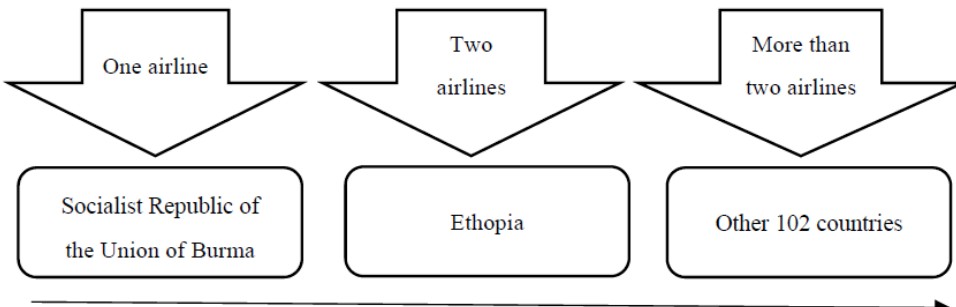

**Figure 2.** The analysis of the right to designate airlines between the Commonwealth of Australia and other countries.

The provision on the exchange of statistics may be mandatory, upon request or required only in cases of disputes overcapacity. The analysis of this regime in all agreements illustrated that the aeronautical authorities of Australia shall supply to the aeronautical authorities of the other contracting party, upon request, such periodic or other statements of statistics as may be reasonably required for the purpose of reviewing the capacity provided on the agreed services by the designated airlines. The statements shall include information relating to the amount of traffic carried by those airlines on the agreed services to and from the territory of the other contracting party, including the origin and destination of the traffic.

The analysis of cooperative arrangements that present a provision for entering into cooperative marketing arrangements, such as blocked-space and code-sharing, illustrated the designated airlines of the parties may enter into code-sharing arrangements with any other airline (Figure 3).

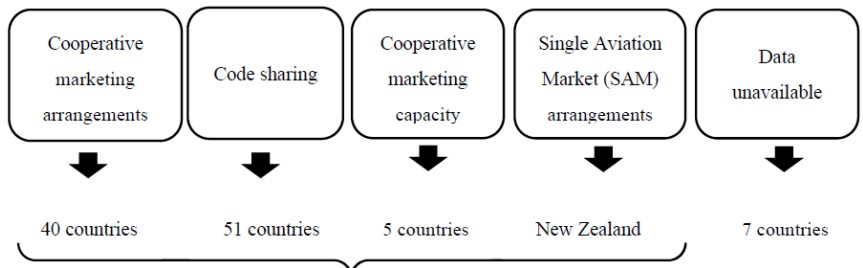

**Figure 3.** The analysis of cooperative arrangements between the Commonwealth of Australia and other countries.

Services under the agreement on any route or sector of a route may carry the designated airline's code in addition to that of the carrier operating the flight as though those services were its own. The designated airline may be required to have the authority to exercise traffic rights over the whole of the route and the other airline may be required to have the authority to exercise traffic rights over the sector or route segment.

The tariff approval regime which governs the approval of the pricing of services between the countries presents that 22 air service agreements include the most restrictive dual approval regime. The free pricing as the most liberal regime presented in 23 agreements. It is interesting to notice that the most restrictive and liberal regimes are usually the most frequent, as shown in Figure 4. Similarly, substantial ownership (34) and the principal place of business (24) are included in most of the agreements. The data analysis of these 2 indicators based on analysis of 58 air service agreements (Figure 4).

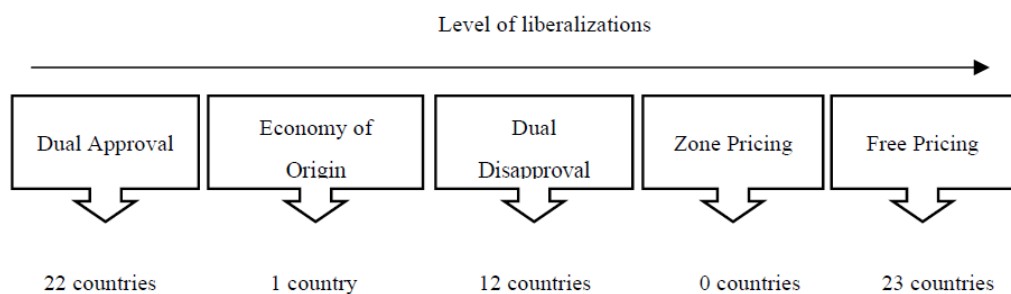

**Figure 4.** The analysis of tariff approval in bilateral agreements between the Commonwealth of Australia and other countries relating to air service.

Analysis of the capacity indicator that shows capacity is agreed prior to the service commencement in 103 air service agreements. Frequency, passenger capacity and aircraft type that depend on destinations should be discussed in 41 cases. For instance, the bilateral agreement between Australia and Canada mentioned that the maximum capacity is 7908 seats per week in each direction to/from Sydney, Melbourne (including Avalon), Brisbane and Perth and unrestricted capacity to/from all other points in Australia. Moreover, designated airlines may exercise full fifth freedom traffic rights at the following intermediate points: Honolulu, Fiji, Tahiti, San Francisco and a point to be agreed. The 14 air service bilateral agreements include passenger capacity limitation based on the maximum number of seats.

For instance, it allowed a maximum 2800 seats per week with any type of aircraft in any configuration between Australia and Netherlands, 2800 seats per week between Australia and Norway, 120 seats between Australia and Samoa and 2800 seats per week Australia and Sweden. From the analysis results of passenger capacity in bilateral agreements between the Commonwealth of Australia and other countries relating to air service, seven different types of predetermination capacity were identified (Figure 5). The 16 bilateral agreements include the information where passenger capacity limits the number of seats per week. For instance, the available capacity between Australia and Cook Island is 1884 seats per week. The passenger capacity entitlements are 2800 seats per week between Australia and Denmark. The other 21 air service agreements, which include the capacity limitation, depend on flight frequency. In the Hong Kong case, it allows a maximum of 38 flights per week between Sydney, Melbourne, Brisbane and Perth and Hong Kong. As well as there being no limit on the number of frequencies that may be operated for passenger services between all points in Australia other than Sydney, Melbourne, Brisbane and Perth and Hong Kong. The agreement between Australia and Malta allows the Australian designated airlines to be entitled in total to operate a maximum of three frequencies per week with any aircraft having a capacity up to that of a B747 aircraft. In the case of agreement with France, the restriction is based on three different routes. In the first route, the capacity limitation depends on units (3.0 units of capacity per week in each direction). In the second route, it allows seven weekly flights in each direction using any aircraft type, and in the last case, the passenger capacity limits of the number of seats (1356 seats per week in each direction).

The biggest number of agreements (33 agreements) include the frequency, capacity and aircraft type (a total of seven services each way each week with any aircraft type) for services to and from Sydney, Melbourne, Brisbane and Perth and the unrestricted frequency, capacity and aircraft type for services to and from all points in Australia other than Sydney, Melbourne, Brisbane and Perth. The agreements between Australia and Bahrain, China, Fiji, New Zealand, Singapore, Switzerland, United Arab Emirates, United Kingdom, United States of America are the most liberal and allow designated Australian airlines to determine the frequency and capacity of services operated.

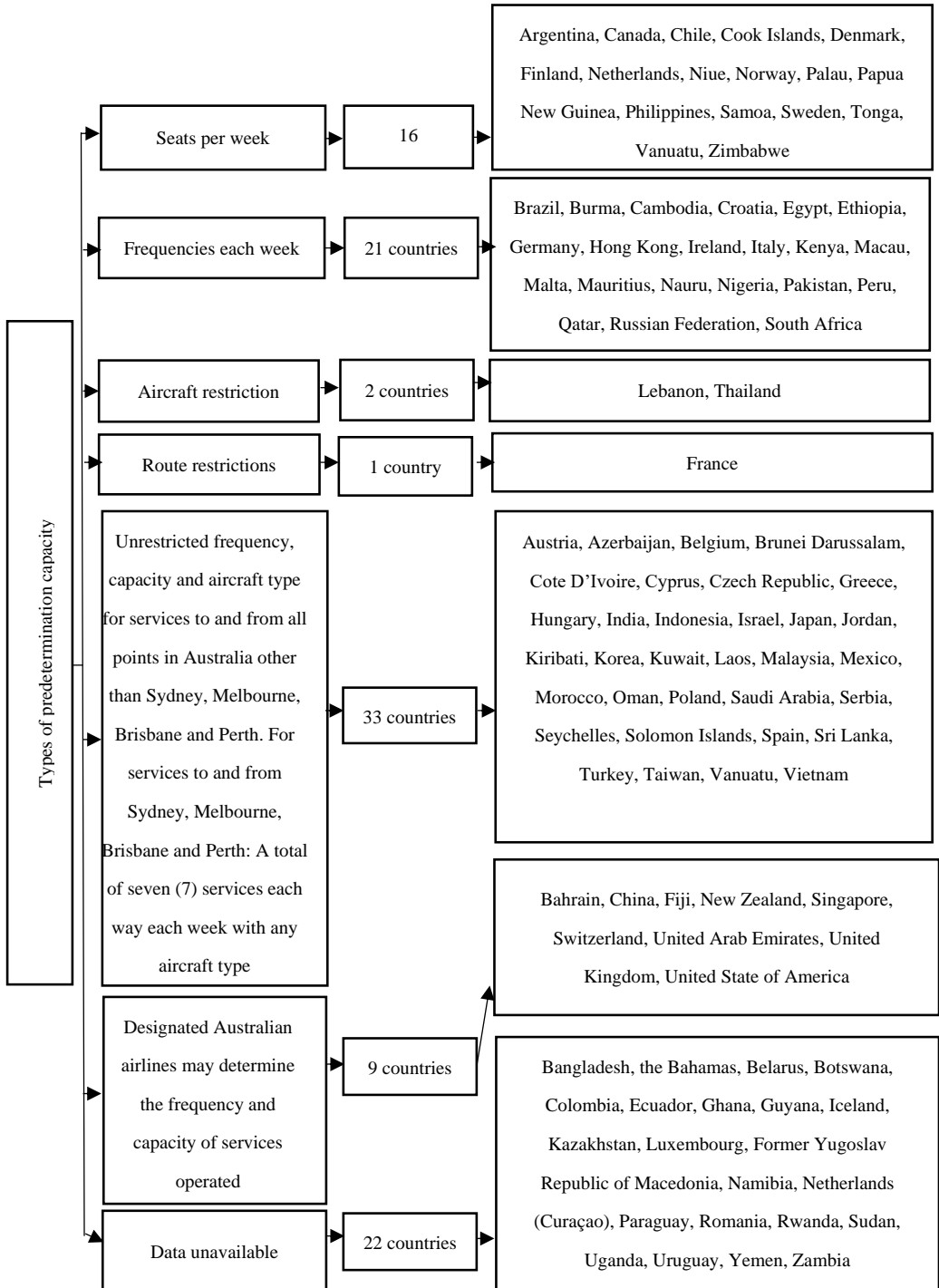

**Figure 5.** Analysis of passenger capacity in bilateral agreements between the Commonwealth of Australia and other countries relating air service.

From the analysis of passenger capacity, six types of predetermination capacity were identified. The passenger capacity restrictions are based on the number of seats, frequencies, routes and types of aircraft, as shown in Figure 5. For instance, available capacity in in bilateral agreements between the Commonwealth of Australia and France relating air service depends on routes. There are 3 variates: 1356 seats per week in each direction, 3.0 units of capacity per week in each direction and 7 weekly frequencies in each direction using any aircraft type. The biggest group of countries (33 partners) have unrestricted frequency, capacity and aircraft type for services to and from all points in Australia other

than Sydney, Melbourne, Brisbane and Perth. Moreover, there are a maximum total of seven services each way each week with any aircraft type for services to and from Sydney, Melbourne, Brisbane and Perth. The highest level of passenger capacity liberalization is with countries, such as Bahrain, China, Fiji, New Zealand, Singapore, Switzerland, United Arab Emirates, United Kingdom and United State of America.

Based on the above analyses, the key elements and factors of the passenger capacity in the Australian Bilateral Air Services Agreements have been identified. Four types of variables in the air service agreements have been analysed, and the special cases in each type have also been pointed out. In the right to designate airlines, most countries can designate multiple airlines in their air agreements with Australia. Moreover, in the cooperative arrangements, the major of countries have been allowed to cooperate with the other parties in the agreement, which contains the "cooperative marketing agreements" and the "code sharing" as the most frequently granted types of rights. Furthermore, for tariffs in the 103 air service agreements with Australia, except for some missing information countries, the numbers of countries that have been granted "Dual approval" and "Free pricing" have occupied the largest place. At last, in passenger capacity, the percentage of countries that have been granted "unrestricted frequency, capacity and aircraft type" is 31.7%, which occupies the first place. Besides, the percentage of countries that have been restricted by the "frequencies per week" is 20.2%, which occupies the second place. For all types, the air agreement between Australia and New Zealand is the most special one as it has SAM airlines. Therefore, the analysis in this section identifies the key features of the Australian air service agreements from designated airlines, cooperative arrangements, tariffs and passenger capacity.

### 4.2. Analysis of Air Cargo Capacity in Australia

From the analysis results, five groups of relation to frequency, capacity or aircraft type for dedicated cargo services were identified in bilateral agreements between the Commonwealth of Australia and other countries relating to air service (Figure 6).

The first group includes countries with tonnes per week cargo capacity limitation, such as Fiji (140 tonnes per week), Palau (150 tonnes each way each week), Papua New Guinea (77.5 tonnes per week in each direction) and Zimbabwe (100 tonnes per week with dedicated freight aircraft). The second group includes 14 countries with weekly cargo services limitation. For instance, airlines from Argentina, Bangladesh, Belgium, Brazil, Ethiopia, Israel, Italy and Vietnam can operate only maximum seven times per week in total with any aircraft type. The cargo capacity for Cambodian airlines is limited to five flights per week and operators from Macau and Saudi Arabia can do up to three round trip flights per week. Two cargo flights per week are allowed for all designed airlines from Netherlands and one weekly cargo flight for operators from Egypt and Burma.

The following three groups are more liberal because designated airline or airlines may determine the frequency, capacity and aircraft type to be operated or there are no limitations in relation to frequency, capacity or aircraft type for dedicated cargo services. There is no limitation on the number of frequencies for all cargo services between Australia and Hong Kong. However, designated airlines of Australia may at their discretion freely convert and reconvert capacity for the operation of passenger services and all-cargo services between Hong Kong and Sydney, Melbourne, Brisbane and Perth on the basis of one passenger frequency for one all-cargo frequency or vice versa. No cargo traffic may be uplifted at an intermediate point and discharged at a point in the territory of the other country. However, designated airlines may uplift traffic at points in their territory for discharge at an intermediate point. Moreover, there is no restriction on the operation of all-cargo services, with any type of aircraft, between Indonesian cities Jakarta, Medan, Surabaya, Denpasar and Makassar and any points in Australia. In operating services between other points in Indonesia and points in Australia, Australia's designated airlines may operate three all cargo services per week in each direction with any aircraft type. In conclusion, cargo capacity analysis illustrates that the level of liberalization is high, but the air services

agreements between Australia and other countries in the first and second cargo capacity groups should be revised to reflect the increase in air cargo traffic during the COVID-19 period (Figure 7).

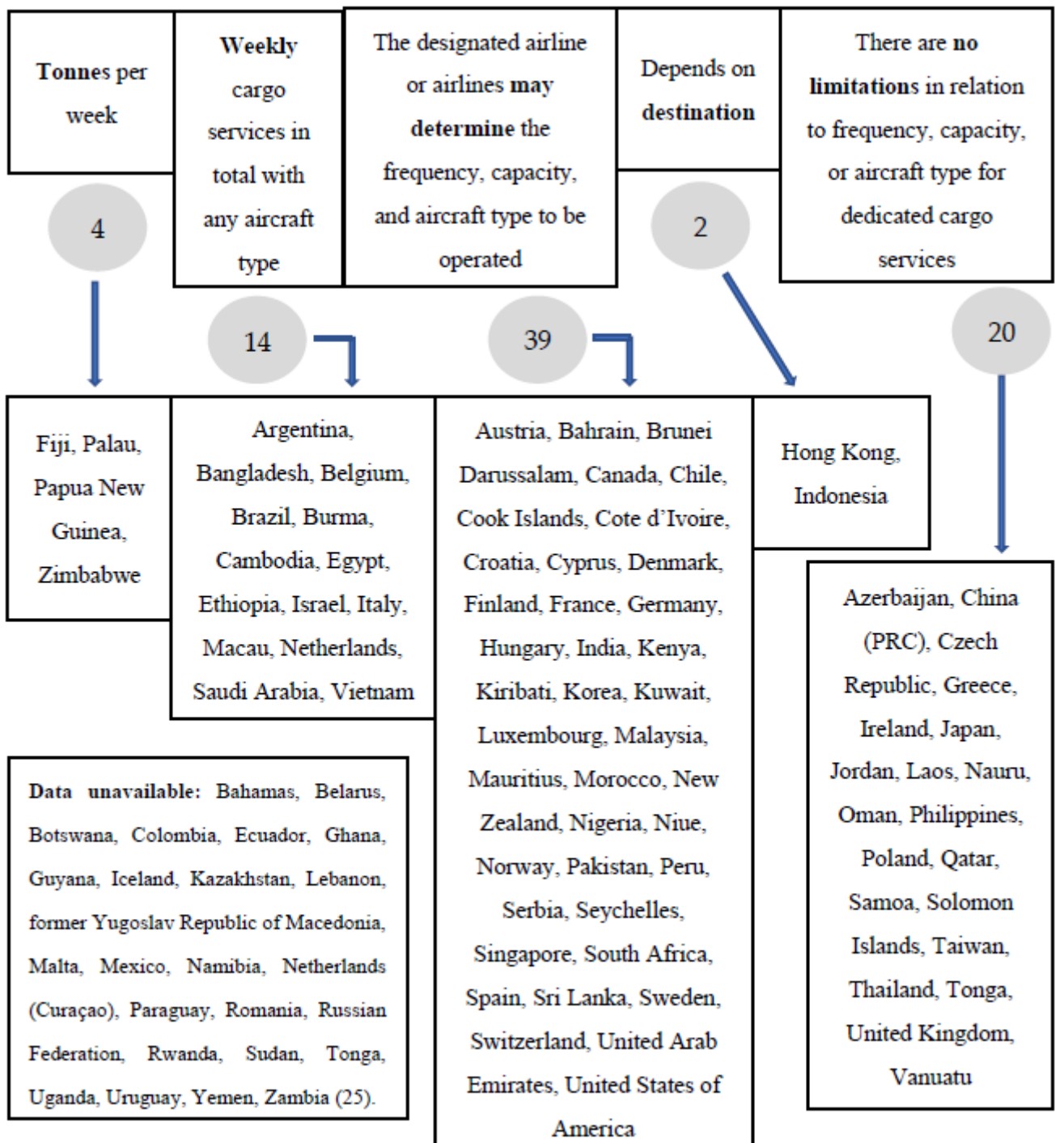

**Figure 6.** Analysis of cargo capacity in bilateral agreements between the Commonwealth of Australia and other countries relating to air service.

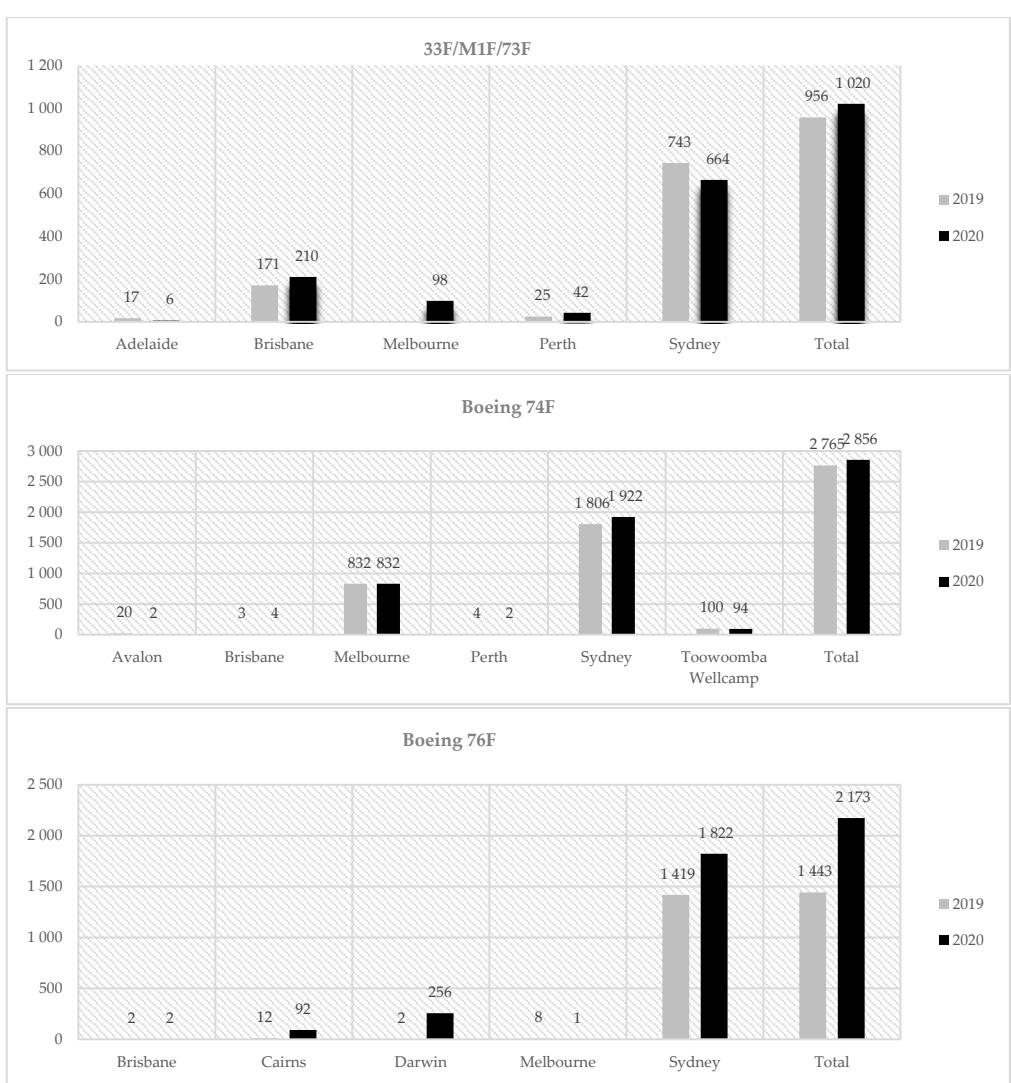

**Figure 7.** Total aircraft movement at Australian international airports by aircraft types (freighters).

From the analyses results, there are three special cases that are worth discussing. The most outstanding country in the Australian Air Liberalization Agreements is New Zealand, which has the highest level of liberalization with Australia. The type of grant of right mentioned in this agreement is the Single Aviation Market (SAM), this type of right allows both parties' designated and not designated airlines to fly between any points in and beyond the other party's territory. What makes the SAM special is that SAM airlines can serve the domestic traffic markets in both two countries, this also distinguishes New Zealand from other countries in the Australian Air Agreement. Besides, the capacity in the Air agreement between Australia and New Zealand is free determination, which allows each party to freely determine their flight capacity and frequency based on the market demand. In order to ensure the greatest extent of implementation of the agreement, each party should not restrict the other party's operation in their market. Moreover, the tariff approval between Australia and New Zealand is also at the highest level; the agreement shows that both two countries can determine the tariff by themselves without filing with the other party's aeronautical authorities. However, these two countries should consider the commercial situations they have as the highest level of freedom in this sector. With the right of principal place of business in withholding, the number of SAM airlines should not be limited and can be jointly approved by both of the countries. Although most countries hold the right to designate more than one airline and flight route in the agreement, New Zealand still gets more freedom since they can designate as many airlines as they want. As mentioned

above, fair, flexible, and free determination are the key elements in the Australia–New Zealand Air Service Agreements. SAM promotes the development of the aviation market in both the two countries. Compared with other countries in the Australia Air Service Liberalization Agreements, New Zealand has the highest freedom to conduct its operating activities and other sales activities in the Australian market and vice versa. All in all, the agreement between Australia and New Zealand has a high degree of free determination. Some rights and priorities have been shared between the Australia and New Zealand aviation markets. Correspondingly, the restrictions and limitations between these two countries' air agreement are also the least in all the Australian air service liberalization agreements. Thus, the air agreement between Australia and New Zealand is unique in all Australian air service agreements.

Another special case in Australian Air Liberalization Agreements is the United Arab Emirates, which also ranked at a high position compared with other countries. One of the factors that causes the particularity of the United Arab Emirates in this agreement is its geographic location and political position. The United Arab Emirates is one of the countries that consist of the Arab states, it is located at the eastern end of the Arabian Peninsula and is surrounded by other Arab countries. However, compared with other Arab countries, the United Arab Emirates has the highest level of liberalization in Australian Air Agreements. Different from New Zealand, the agreement with the United Arab Emirates limited the grant of the right to only designated airlines in agreed routes. However, the right in this agreement is still higher than the agreements with a big part of other countries since it allows both of the parties to fly via designated intermediate points to the points beyond. Besides, it has been mentioned in the agreement that each party should maintain a close relationship with another party, which suggests market demand and economic environment should be put at the first positions when they determine their capacity and tariff by themselves. Moreover, the United Arab Emirates holds substantial ownership and effective control in the withholding sector, which requires each part to comply with all the regulations and laws to satisfy the other party. The agreement between the United Arab Emirates and Australia also allows these two parties to designate the airlines and flight routes by themselves, however, written notifications should be provided to the other party. Fair and equality operations have been mentioned repetitively in the Cooperative Arrangements sector, these are also the cores in the Australia–United Arab Emirates Air Service Agreements that should be strictly complied by the two parties. To summarize, the key elements in the agreement between Australia and the United Arab Emirates are appropriate, flexible and fair. The rights and agreement level of the agreement with the United Arab Emirates is the highest compared with other Arab countries, however, there are still a lot of limitations and restrictions.

In Australian Air Liberalization Agreements, Singapore is also a special case that should be paid attention to. Singapore has a high level of liberalization in the sectors of capacity, withholding, designation and cooperative arrangements, however, there are more restrictions in the Tariff Approval sector if compared with other same-level countries. Thus, the total score of the air liberalization between Australia and Singapore is ranked at a middle level. Similar to the situation in the United Arab Emirates, Singapore can also operate its designated airlines on the agreed routes. Moreover, the flights and airlines' capacity should be determined under the market needs and also consider the other party's operation needs. In this sector, being fair and eliminating the bad impact in operation has been mentioned and highlighted. Singapore holds the same types of rights as the United Arab Emirates in the withholding and designated sectors, however, all rights and relevant changes should be agreed upon by both of the parties in the agreement between Australia and Singapore. Different from New Zealand and the United Arab Emirates, the set tariffs should be agreed upon by both of the two countries in the agreement, which is dual approval. Besides, close cooperation is the keyword in the cooperative arrangements sector, several actions can be taken to ensure both parties' fair operation in the other party's territory. All in all, partial rights in the agreement of Australia and Singapore are at a high

level. Being fair and mutually agreed are the two most important elements in the agreement. However, compared with the United Arab Emirates, the conditions in Australia–Singapore Air Service Liberalization Agreements have more restrictions.

From the above analysis, different countries' cargo capacity in Australian Air Liberalization Agreements has been identified from five types of predetermination capacity. In these agreements, the number of countries that have been restricted by the number and types of the designated airlines occupied the first place. Besides, the percentage of countries that have no limitation in the cargo capacity in Australian Air Liberalization Agreements is about 25.3%, which occupies second place. The type that occupies the last place is "depending on destination", and the countries that have been granted this type of cargo capacity are Indonesia and Hong Kong. Besides, based on the different countries' cargo capacity and other variables in Australian Air Liberalization Agreements, the three most special cases have been analysed to identify how Australia builds different Air agreements based on the different conditions in other countries. As mentioned above, New Zealand has been identified as the most special country in Australian Air Liberalization Agreements. The Single Aviation Market (SAM) enables New Zealand and Australia to have the highest priority in the other parties' aviation market. This high level of air liberalization was caused by both political issues and geographic reasons. However, based on Australia's prudence concept, the level of air liberalization has been weakened in the agreements between Australia and some Asian countries. However, the economic factor and other factors result in a high level of capacity and cooperation in the Air agreements between Australia with these countries, which can be seen in the cases of Singapore and the United Arab Emirates. Therefore, when analysing the level of air liberalization in Australian Air Liberalization Agreements, other factors should also be considered to identify how to improve the performance of the Australian aviation industry reasonably.

*4.3. Air Liberalization Index*

Air Liberalization Index can be built to provide an indication of the overall degree of liberalization introduced by a certain air service agreement. As shown in Table 1, the four ALIs indicate a low level of liberalization of air services. Table 2 above showed that restrictive regimes are very frequent in the design of ASAs. Besides, the unavailable data and limited access to the bilateral agreement do not allow the assessment of the overall degree of liberalization of ASAs. All information used in the section has been sourced from the Department of Infrastructure, Transport, Regional Development and Communications of Australia (Australian government, 2021). Based on the information in the Air Agreements of different countries, the points of each country have been allocated, and then calculated according to each standard of the Air Liberalization Index.

Approximately 43 per cent of ASAs presents an ali_standard (FA_index) have six points. Very few ASAs introduce an intermediate degree of liberalization (in the range 10–16 for the ali_standard). A high degree of liberalization of the aviation market (measured by an ali_standard in the range 24–37) is reached only in two countries: Macau and New Zealand. This is mainly due the signed the Single Aviation Market agreement with New Zealand.

Within the data available countries, approximately 43% of the countries' air agreements with Australia can be identified as Type C, which has a higher restriction on designation rights. Moreover, the number of countries in Type o have occupied the second largest place, which is about 29.3%. The most special case is New Zealand, which is the only country in Type G. Among all countries, New Zealand's air agreement with Australia has the highest liberalization. Therefore, since only one country can be classified in Type G, the air agreements between Australia and other countries are not flexible.

**Table 1.** Calculation of Air Liberalization Index between Australia and other countries.

| Country | Date | Direct Service March 2020 | ALI Standard | ALI 5th+ | ALI OWN+ | ALI DES+ | Type |
|---|---|---|---|---|---|---|---|
| Argentina | 25 May 2012 | No | 6 | 12 | 5 | 5.5 | C |
| Austria | 12 August 2009 | No | 10 | 15.5 | 8,5 | 9 | D |
| Bahrain | 18 July 2003 | No | 19 | 23 | 16 | 21 | o |
| Belgium | 21 February 2011 | No | 6 | 12 | 5 | 5.5 | C |
| Brazil | 24 July 2008 | No | 6 | 12 | 5 | 5.5 | C |
| Brunei Darussalam | 28 October 2015 | Yes | 6 | 12 | 5 | 5.5 | C |
| Burma | 29 October 2015 | No | 6 | 12 | 5 | 5.5 | C |
| Canada | 21 March 2017 | Yes | 10 | 15.5 | 8,5 | 13 | E |
| Chile | 17 June 2019 | Yes | 10 | 15.5 | 8,5 | 13 | E |
| China (PRC) | 28 February 2017 | Yes | 6 | 12 | 5 | 5.5 | C |
| Cook Islands | 2 April 2019 | Yes | 18 | 15.5 | 22.5 | 20.5 | o |
| Croatia | 16 March 2007 | No | 6 | 12 | 5 | 5.5 | C |
| Czech Republic | 8 August 2005 | No | 6 | 12 | 5 | 5.5 | C |
| Egypt | 17 December 2012 | No | 12 | 17 | 10 | 14.5 | i |
| France | 27 May 2020 | No | 14 | 19 | 12 | 16.5 | F |
| Germany | 2 May 2013 | No | 10 | 15.5 | 8,5 | 13 | E |
| Greece | 9 January 2018 | No | 6 | 12 | 5 | 5.5 | C |
| Hong Kong | 25 March 2021 | Yes | 15 | 13 | 20 | 17.5 | o |
| Hungary | 19 December 2006 | No | 6 | 12 | 5 | 5.5 | C |
| India | 26 June 2018 | Yes | 10 | 15.5 | 8,5 | 13 | E |
| Indonesia | 10 February 2020 | Yes | 15 | 13 | 20 | 17.5 | o |
| Ireland | 8 August 2005 | No | 10 | 15.5 | 8,5 | 9 | D |
| Israel | 8 August 2017 | No | 19 | 23 | 16 | 21 | o |
| Italy | 9 January 2018 | No | 6 | 12 | 5 | 5.5 | C |
| Japan | 28 October 2019 | Yes | 8 | 13.5 | 6.5 | 7 | i |
| Korea, Rep of | 6 February 2019 | Yes | 10 | 15.5 | 8,5 | 13 | E |
| Kuwait | 23 February 2017 | No | 6 | 12 | 5 | 5.5 | C |
| Laos | 23 February 2017 | No | 6 | 12 | 5 | 5.5 | C |
| Lebanon | 9 June 2000 | No | 16 | 20.5 | 13.5 | 18.5 | o |
| Macau | 13 December 2011 | No | 24 | 27.5 | 27.5 | 26 | o |
| Malaysia | 9 January 2018 | Yes | 6 | 12 | 5 | 5.5 | C |
| Malta | 18 July 2017 | No | 16 | 20.5 | 13.5 | 18.5 | o |
| Mauritius | 18 October 2019 | Yes | 6 | 12 | 5 | 5.5 | C |
| Mexico | 14 April 2005 | No | 6 | 12 | 5 | 5.5 | C |
| Nauru | 9 March 2016 | Yes | 6 | 12 | 5 | 5.5 | C |
| Netherlands | 23 November 2012 | No | 10 | 15.5 | 8,5 | 13 | E |
| New Zealand | 28 February 2017 | Yes | 37 | 32 | 39 | 38 | G |
| Palau | 19 April 2013 | No | 6 | 12 | 5 | 5.5 | C |
| Papua New Guinea | 26 February 2019 | Yes | 6 | 12 | 5 | 5.5 | C |
| Philippines | 5 October 2018 | Yes | 6 | 12 | 5 | 5.5 | C |
| Poland | 1 July 2003 | No | 6 | 12 | 5 | 5.5 | C |
| Russian Federation | 23 February 1999 | No | 16 | 20.5 | 13.5 | 18.5 | o |
| Samoa | 11 August 2017 | Yes | 16 | 20.5 | 13.5 | 18.5 | o |
| Serbia | 23 February 2017 | No | 6 | 12 | 5 | 5.5 | C |

**Table 1.** *Cont.*

| Country | Date | Direct Service March 2020 | ALI Standard | ALI 5th+ | ALI OWN+ | ALI DES+ | Type |
|---------|------|---------------------------|--------------|----------|----------|----------|------|
| Singapore | 26 October 2015 | Yes | 6 | 12 | 5 | 5.5 | C |
| Solomon Islands | 23 February 2017 | Yes | 19 | 23 | 16 | 21 | o |
| South Africa | 27 April 2011 | Yes | 16 | 20.5 | 13.5 | 18.5 | o |
| Spain | 16 March 2007 | No | 6 | 12 | 5 | 5.5 | C |
| Sri Lanka | 17 December 2018 | Yes | 10 | 15.5 | 8,5 | 13 | E |
| Switzerland | 12 June 2012 | No | 6 | 12 | 5 | 5.5 | C |
| Thailand | 3 May 2016 | Yes | 10 | 15.5 | 8.5 | 9 | D |
| Tonga | 29 September 2008 | Yes | 19 | 23 | 16 | 21 | o |
| Turkey | 23 February 2017 | No | 19 | 23 | 16 | 21 | o |
| United Arab Emirates | 10 April 2017 | Yes | 19 | 23 | 16 | 21 | o |
| United Kingdom | 26 October 2006 | Yes | 14 | 19 | 12 | 16.5 | F |
| United States of America | 28 February 2017 | Yes | 8 | 13.5 | 6.5 | 7 | i |
| Vanuatu | 21 March 2017 | Yes | 19 | 23 | 16 | 21 | o |
| Vietnam | 21 March 2017 | Yes | 16 | 20.5 | 13.5 | 18.5 | o |
| Data unavailable | Azerbaijan, Bahamas, Bangladesh, Belarus, Botswana, Cambodia, Colombia, Cote d'Ivoire, Cyprus, Denmark, Ecuador, Ethiopia, Fiji, Finland, Ghana, Guyana, Iceland, Jordan, Kazakhstan, Kenya, Kiribati, Luxembourg, former Yugoslav Republic of Macedonia, Morocco, Namibia, Netherlands (Curaçao), Nigeria, Niue, Norway, Oman, Pakistan, Paraguay, Peru, Qatar, Romania, Rwanda, Saudi Arabia, Seychelles, Sudan, Sweden, Taiwan, Uganda, Uruguay, Yemen, Zambia, Zimbabwe | | | | | | |

**Table 2.** Grouping of countries depends on type of Air Service Agreement.

| Type | # | Countries |
|------|---|-----------|
| A | 0 | none |
| B | 0 | none |
| C | 25 | Argentina, Belgium, Brazil, Brunei Darussalam, Burma, China (PRC), Croatia, Czech Republic, Greece, Hungary, Italy, Kuwait, Laos, Malaysia, Mauritius, Mexico, Nauru, Palau, Papua New Guinea, Philippines, Poland, Serbia, Singapore, Spain, Switzerland |
| D | 3 | Austria, Ireland, Thailand |
| E | 7 | Canada, Chile, India, Korea, Rep of, Netherlands, Sri Lanka, United Kingdom |
| F | 2 | France, Germany |
| G | 1 | New Zealand |
| i | 3 | Egypt, Japan, United States of America |
| o | 17 | Bahrain, Cook Islands, Hong Kong, Indonesia, Israel, Lebanon, Macau, Malta, Russian Federation, Samoa, Solomon Islands, South Africa, Tonga, Turkey, United Arab Emirates, Vanuatu, Vietnam |

From the above figures and analyses, the final score of each country in the Australian Air Liberalization Agreements has been calculated. As introduced in the Methodology section, the Air Liberalization Index was developed by WTO Secretariat [35]. Besides, the specific criteria of the index have been provided in Appendix A Table A2. To ensure the reliability and comprehensiveness of the results, the air liberalization scores of Australia with other countries have been calculated based on all the four standards of the Air Liberalization Index. It can be seen from the above tables that New Zealand has been ranked first place by using all four standards, followed by the United Kingdom, which scored as the second highest. Besides, Belgium has been ranked third place using most of the standards, and it has the fourth-highest score by using the 5th+ standard to calculate

the Air Liberalization Index. Moreover, the laggards were most of the Asian countries and some of the European countries. Besides, some countries from Africa and most of the third world countries were at the very bottom of the list. Besides, about half of the countries do not have direct service with Australia (data until March 2020). Therefore, the results support that the level of Australian Air Liberalization is not high enough, and there is still a large room for development. Besides, since the travel restrictions have been released in early 2020, it is essential to provide higher air liberalization and generate more opportunities for the aviation industry to improve its performance. In this situation, the analysis of Australian Air Liberalization enables the policymakers to develop more effective and realistic policies to help the industry bounce back as before.

## 5. Conclusions and Recommendations

In this paper, an analysis of the air bilateral agreements between Australia and other countries was conducted in order to identify the level of air service liberalization and main market features (i.e., designation, withholding, tariffs, capacity, traffic rights, absence of exchange of statistics, allowance of cooperative arrangements) that should be reviewed post the COVID-19 pandemic.

This paper uses the ALI index to assess the degree of liberalization of the aviation market resulting from bilateral air service agreements. A total of 43% of ASAs presents a low ali_standard of 6 points. Very few ASAs introduce an intermediate degree of liberalization (in the 10–16 range of the ali_standard). A high degree of liberalization of the aviation market (measured by an ali_standard in the range 24–37) is achieved only in two countries, Macau and New Zealand. This is mainly due the signed the Single Aviation Market agreement with New Zealand.

From the data and our analysis, we found that the degree of liberalization of the aviation market is quite low. Supporting this conclusion includes the analysis on grants showing airline operations based on the four freedoms of the air. Australia only has a Single Aviation Market with New Zealand that allows their designated and non-designated airlines to fly between any points in and beyond each other's territories. Second, the analyses of cooperative arrangements show that the designated airlines may enter into code-share agreements with other air carriers. A designated airline may be an operating airline, a marketing (non-operating) airline or both. It was found that Australia's borders are opened for multiple airlines from more than 100 countries (exceptions are agreements with the Socialist Republic of the Union of Burma and Ethiopia). Third, the revision of the capacity indicator shows capacity is agreed upon prior to the service commencement in 103 air service agreements. From the analysis of passenger capacity, six types of predetermination capacity were identified. Lastly, it was found that the volume of cargo transportation between Australia and other countries increased during the COVID-19 pandemic.

To mitigate some of the negative impacts caused by the COVID-19 pandemic to the aviation industry, it is recommended for Australia to further open up the level of freedoms in the BASAs with highly developed countries and regions to stimulate and increase market interactions. Another recommendation is to increase the flexibility in the agreements to encourage more foreign airlines to participate in the code-sharing market. A large number of BASAs between Australia with other countries have restricted designated airlines and routes, and the capacity is also limited. This was intended as a protection for both stakeholder's operational needs before the COVID-19 outbreak. However, with the global aviation market suffering an unprecedented and continuous financial loss through the COVID-19 pandemic, policies of this nature should be reviewed with the view to loosen the restrictions. Each stakeholder can develop their own strategies in a more flexible way with the possibility of more airlines and flights to meet the needs of a changing market.

It is worth also highlighting here that a limitation of this research is due to the non-availability of some of the agreements (as listed in the data) to the authors at the time of this study. In conclusion, this study provides an insight into the current status of Australia's BASAs with recommendations of actions to stem and reverse the negative impacts caused

by the COVID-19 pandemic to the aviation industry in a post COVID-19 environment. It is envisioned that this study will add to the current body of knowledge given that there is only a limited number of previous research done on BASAs liberalization in Australia.

**Author Contributions:** Conceptualization, I.H., R.C.K.Y., W.J.C.V. and J.L.; methodology, I.H. and J.L.; validation, I.H., R.C.K.Y., W.J.C.V. and J.L.; formal analysis, I.H., R.C.K.Y., W.J.C.V. and J.L.; investigation, I.H.; resources J.L.; data curation I.H. and J.L.; writing—original draft preparation I.H.; writing—I.H., R.C.K.Y., W.J.C.V. and J.L.; visualization, I.H., R.C.K.Y., W.J.C.V. and J.L.; supervision, R.C.K.Y. All authors have read and agreed to the published version of the manuscript.

**Funding:** RMIT University. RI-00101-001. STEM > SoE Support A&A Iryna Heiets.

**Institutional Review Board Statement:** Not applicable.

**Informed Consent Statement:** Not applicable.

**Data Availability Statement:** Not applicable.

**Conflicts of Interest:** The authors declare no conflict of interest.

**Appendix A**

**Table A1.** Air Liberalization Index weighting systems.

| Element | Air Liberalisation Index | | | |
| --- | --- | --- | --- | --- |
| | **Standard** | **5th+** | **OWN+** | **DES+** |
| GRANT OF RIGHTS | | | | |
| Fifth Freedom | 6 | 12 | 5 | 5.5 |
| Seventh Freedom | 6 | 5 | 5 | 5.5 |
| Cabotage | 6 | 5 | 5 | 5.5 |
| CAPACITY | | | | |
| Predetermination | 0 | 0 | 0 | 0 |
| Other restrictive | 2 | 1.5 | 1.5 | 1.5 |
| Bermuda I | 4 | 3.5 | 3.5 | 3.5 |
| Other liberal | 6 | 5 | 5 | 5.5 |
| Free Determination | 8 | 7 | 7 | 7.5 |
| TARIFFS | | | | |
| Dual Approval | 0 | 0 | 0 | 0 |
| Economy of Origin | 3 | 2.5 | 2.5 | 2.5 |
| Dual Disapproval | 6 | 5 | 5 | 5.5 |
| Zone Pricing | 8 / 4 / 7 | 7 / 3.5 / 6 | 7 / 3.5 / 6 | 7.5 / 3.5 / 6.5 |
| Free Pricing | 8 | 7 | 7 | 7.5 |
| WITHHOLDING | | | | |
| Substantial Ownership and Effective Control | 0 | 0 | 0 | 0 |
| Community of Interest | 4 | 3.5 | 7 | 3.5 |
| Principal Place of Business | 8 | 7 | 14 | 7.5 |
| DESIGNATION | | | | |
| Single Designation | 0 | 0 | 0 | 0 |
| Multiple Designation | 4 | 3.5 | 3.5 | 7.5 |
| STATISTICS | | | | |
| Exchange of Statistics | 0 | 0 | 0 | 0 |
| No exchange of Statistics | 1 | 1 | 1 | 1 |
| COOPERATIVE ARRANGEMENTS | | | | |
| Not allowed | 0 | 0 | 0 | 0 |
| Allowed | 3 | 2.5 | 2.5 | 2.5 |
| TOTAL | 50 | 50 | 50 | 50 |

**Table A2.** Types of Air Services Agreements.

| Type | Freedoms | Designation | Withholding/Ownership | Tariffs | Capacity |
|---|---|---|---|---|---|
| A | 3rd and 4th | Single designation | Substantive ownership and effective control | Double approval | Pre-determination |
| B | 3rd and 4th | Multi-designation | Substantive ownership and effective control | Double approval | Pre-determination |
| C | 3rd, 4th, 5th | Single designation | Substantive ownership and effective control | Double approval | Pre-determination |
| D | 3rd, 4th, 5th | Single designation | Substantive ownership and effective control | Double approval | Bermuda I |
| E | 3rd, 4th, 5th | Multi-designation | Substantive ownership and effective control | Double approval | Pre-determination |
| F | 3rd, 4th, 5th | Multi-designation | Substantive ownership and effective control | Double approval | Bermuda I |
| G | 3rd, 4th, 5th | Multi-designation | Substantive ownership and effective control or Community of interest or Principal place of business | Free pricing or Double disapproval | Free determination |
| i Incomplete ICAO coding O All other combinations | If either: | | n/a | n/a | other |

Note: n/a denotes the non-availability of the relevant information.

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
