# Peer review of "Analysis of Bilateral Air Services Agreement Liberalization in Australia"

_aerospace, doi:10.3390/aerospace9070371_

Round 1

Reviewer 1 Report

This paper reported an analysis of the air bilateral agreements between Australia and other countries and regions was conducted in order to identify the level of air service liberalization and main market features that should be reviewed post the COVID-19 pandemic.

The paper is well prepared but with a style of commercial report rather than a scientific paper. The topic is interesting for discuss and improve.

It is better to reduce some parts of the text.

Most figures and tables are out of the marge, and it is better to re-design.

It is better to give some math presentation of ALI index for more clear.

Author Response

The authors would like to thank the reviewers for their thorough reviews, including helpful comments and suggestions. This has enabled us to revise our submission to meet the quality requirements associated with publication in Aerospace. Our response to the reviewer comments is given below. The revised manuscript file includes text in red font to denote the corresponding changes to the manuscript, as further discussed below.

Reviewer #1 comments

Comment

Response to reviewer comment

The paper is well prepared but with a style of commercial report rather than a scientific paper. The topic is interesting for discuss and improve.

The paper has been revised to comply with scientific paper standards. In response to this comment as well as several from reviewers #2 and #3 (see response to their comments as given below), the following updates have been included:

1)     The introduction has been revised to clarify the problem statement, the gaps in the state of the art, the objective of this research, and the associated novel contributions.

2)     Section 3 has been updated to clearly describe the research methodology and its constituent elements.

It is better to reduce some parts of the text.

The paper has been updated comprehensively to address the shortcomings identified in the reviewers’ comments. As part of this process, we have critically reviewed the length of the manuscript and have reduced several sections. The revised manuscript has been reduced by 10601 words.

Most figures and tables are out of the marge, and it is better to re-design.

The figures and tables have been revisited and re-designed to ensure they stay within margins while increasing overall figure and table clarity.

It is better to give some math presentation of ALI index for more clear.

As noted previously, Section 3 has been updated to describe the research methodology in detail. As part of this section, the ALI index has been expressed in mathematical terms (lines 225 – 230).

Reviewer 2 Report

I am pleased to read the content of the article. The topic is interesting and topical. Below I present my comments and suggestions:

Abstract

- The authors write: "This paper examines air transport services liberalization, alternative measures of liberalization of the aviation market and the impacts of COVID-19 on airline passenger and freight capacity in Australia." - the purpose of the research is not clearly defined. In fact, you can read up to 3 goals. This is not good.

Introduction

- the authors did not explain properly why they took up this topic, what they want to get, what they strive for through their analysis, why this research is important, etc .;

- and here there is no indication of the goals of the research;

- the introduction should also include the research hypothesis / thesis and research questions - they prove that research is deliberate - this article does not contain this.

Literature review

The literature review was not summarized, the authors did not reach conclusions and what research gaps they identified.

Methodology

I believe that the methodology used is poorly described - i.e. I do not see what specific research methods have been used and why these and not others.

Results

As the research method was not properly selected, the results are not objective but rather intuitive.

Charts 7-10 is of very poor quality. It is not professional to insert a drawing of this quality.

This section is not consistent. This is due to the lack of goals, or the setting of as many as 3 goals. First, the issue of liberalization is described, and then the dynamics of transport, including COVID-19, is very detailed. What is this for?

References

Poor literature. Only 35 items is definitely not enough.

A final note

The article is not yet well prepared for publication. It requires reflection and many changes. I do not recommend it for further procedure.

Author Response

The authors would like to thank the reviewers for their thorough reviews, including helpful comments and suggestions. This has enabled us to revise our submission to meet the quality requirements associated with publication in Aerospace. Our response to the reviewer comments is given below. The revised manuscript file includes text in red font to denote the corresponding changes to the manuscript, as further discussed below.

Comment

Response to reviewer comment

Abstract

- The authors write: "This paper examines air transport services liberalization, alternative measures of liberalization of the aviation market and the impacts of COVID-19 on airline passenger and freight capacity in Australia." - the purpose of the research is not clearly defined. In fact, you can read up to 3 goals. This is not good.

This sentence has been reformulated to clarify the purpose and contributions of this research. The revised formulation is as follows:  this research aims to provide an assessment of the level of air transport services liberalization specific to Australia. Furthermore, it aims to generate recommendations what key market access features of Air Services Agreements should be revised to reflect the changes in air transport characteristics, including the increase in air cargo traffic, during the COVID-19 period

Introduction

- the authors did not explain properly why they took up this topic, what they want to get, what they strive for through their analysis, why this research is important, etc .;

- and here there is no indication of the goals of the research;

- the introduction should also include the research hypothesis / thesis and research questions - they prove that research is deliberate - this article does not contain this.

The paper has been updated comprehensively to address the shortcomings identified in the reviewers’ comments. In particular, the revised introduction now clearly addresses:

·       The problem statement, including the rationale for and impact of the research vis-à-vis academic and practical considerations, has been clarified and now leads organically into the research aims and contributions;

·       The research aims have been explicitly identified and formulated in unambiguous terms;

·       Research questions have been included.

Please refer to the highlighted changes in the introduction section (line 38 – 81) of the revised manuscript.

Literature review

The literature review was not summarized, the authors did not reach conclusions and what research gaps they identified.

A brief subsection has been added to the literature review (Section 2.3, lines 205-220) in which the current state of the art is summarized and relevant research gaps are identified.

Methodology

I believe that the methodology used is poorly described - i.e. I do not see what specific research methods have been used and why these and not others.

Section 3 has been updated to include a clear overview of applicable research methods and the supporting selection rationale. See line 225-230 and 243 onwards for relevant updates.

Results

As the research method was not properly selected, the results are not objective but rather intuitive.

Charts 7-10 is of very poor quality. It is not professional to insert a drawing of this quality.

This section is not consistent. This is due to the lack of goals, or the setting of as many as 3 goals. First, the issue of liberalization is described, and then the dynamics of transport, including COVID-19, is very detailed. What is this for?

Relative to the first point, the updated methodology section now leads more naturally into the results section. In our view, the results section is structured in line with the methodology and follows a systematic process to yield objective insights relative to the research question.

Charts 7 – 10 have been updated.

The structure, narrative and content of the Results section has been thoroughly revised to ensure consistency with the research aims and questions.

References

Poor literature. Only 35 items is definitely not enough.

We respectfully disagree with this comment. The included references are primarily drawn from top-tier journals, with several relevant technical reports adding legislative and industrial perspectives to inform the discussion regarding the topic at hand. In our view (and experience with having authored a considerable quantity of Q1 journal papers and having guest edited several journal special issues), the quantity of references is very much in line with typical journal publications.

Reviewer 3 Report

The article sent for evaluation is a coherent, interesting study relating to the issues related to the liberalization of air transport services, alternative means of liberalizing the aviation market, taking into account the impact of COVID-19. The discussion related to the presented research results is an interesting contribution to the discussion on the liberalization of air transport.

Despite the preliminary opinion defined in this way, I am asking the Authors to refer to the following comments:

1.       p.5 - Methodology - no clear indication of the main purpose of the research - please define it.

2.     p.6. - Results - I propose to change the title to "Research results". In addition, an short introduction to subsections should be added after the chapter title.

3.       p.11 - at the end of subchapter 4.1. Please provide a short comment relating to all the examined sub-areas, and not only to the comments on Figure 5.

4.       pp. 11-19 - 4.2. Analysis of Impact (...) - please provide sources of statistical data and other information referred to by the author (s) in this subsection. Please define the broader end of the subsection relating to its entirety.

5.       p. 19-23 - 4.3. Analysis of Air cargo (...) - please provide the sources of statistical data and other information referred to by the author (s) in this subsection. Please define the broader end of the subsection relating to its entirety.

6.   pp. 23-24 - 4.4. Air Liberalization (…) - No data sources and no final comment. There is no clear definition of the gradation / methodology for awarding points to which the author (s) referred to in Table 1. In my opinion, this issue can be referred to in the methodology section. Moreover, the sentence - line 800 - should read "(...) Table 2 below (...)", and is (...) Table 2 above (...).

7.       pp. 26-27 - Literature: not all items of literature are cited in the study - item 23, 25. Moreover, I would like to ask you to check the compliance of the rules used by the author (s) in the description of the literature with those specified by the Journal.

General remark - please complete the drawings in the whole article with sources in accordance with the applicable rules.

Author Response

The authors would like to thank the reviewers for their thorough reviews, including helpful comments and suggestions. This has enabled us to revise our submission to meet the quality requirements associated with publication in Aerospace. Our response to the reviewer comments is given below. The revised manuscript file includes text in red font to denote the corresponding changes to the manuscript, as further discussed below.

Reviewer #3

Comment

Response to reviewer comment

p.5 - Methodology - no clear indication of the main purpose of the research - please define it.

Well-noted. As mentioned in our reply to reviewer #2, we have updated the introduction comprehensively to address the shortcomings identified in the reviewers’ comments. In particular, the revised introduction now clearly addresses:

·       The problem statement, including the rationale for and impact of the research vis-à-vis academic and practical considerations, has been clarified and now leads organically into the research aims and contributions;

·       The research aims have been explicitly identified and formulated in unambiguous terms;

Please refer to the highlighted changes in the introduction section (line 38 – 81) of the revised manuscript.

p.6. - Results - I propose to change the title to "Research results". In addition, an short introduction to subsections should be added after the chapter title.

Thank you for this suggestion. It has been adopted and the brief introduction of each subsection including expected findings has been provided.

p.11 - at the end of subchapter 4.1. Please provide a short comment relating to all the examined sub-areas, and not only to the comments on Figure 5.

Thank you for this suggestion. A summary of the whole section, including key findings, has been provided at the end of the section.

pp. 11-19 - 4.2. Analysis of Impact (...) - please provide sources of statistical data and other information referred to by the author (s) in this subsection. Please define the broader end of the subsection relating to its entirety.

Thank you for this suggestion. According to the comments and project objective of the entire study, findings and results of the original section 4.2 were removed to make the study to be more consistent and coherent.

p. 19-23 - 4.3. Analysis of Air cargo (...) - please provide the sources of statistical data and other information referred to by the author (s) in this subsection. Please define the broader end of the subsection relating to its entirety.

Thank you for this suggestion. The key sources of this part have been added at the beginning of this section. Besides, a summary of the whole section, including key findings, has been provided at the end of the section.

pp. 23-24 - 4.4. Air Liberalization (…) - No data sources and no final comment. There is no clear definition of the gradation / methodology for awarding points to which the author (s) referred to in Table 1. In my opinion, this issue can be referred to in the methodology section. Moreover, the sentence - line 800 - should read "(...) Table 2 below (...)", and is (...) Table 2 above (...).

Thank you for this suggestion. The following changes have been made to address the comments:

-        Data and information sources have been provided at the first paragraph of this section.

-        Discussions and the final summary of this section have also been provided at the end of the section, and the line 800 has been corrected.

-        Besides, the criteria and specific information of the Air Liberalization Index used in this section have been provided in the Annex 1. As the Methodology part introduced, this index was developed by WTO Secretariat (WTO, 2006). 

pp. 26-27 - Literature: not all items of literature are cited in the study - item 23, 25. Moreover, I would like to ask you to check the compliance of the rules used by the author (s) in the description of the literature with those specified by the Journal.

Line 152

General remark - please complete the drawings in the whole article with sources in accordance with the applicable rules.

Where information has been used from external sources, referencing has been included. Several figures are original artwork and as such not sourced.

Round 2

Reviewer 2 Report

Based on the manuscript I received for re-examination, I make the following comments: a / Abstract - my attention has not been taken into account, there is still a sentence with 3 equivalent goals: "This paper examines air transport services liberalization, alternative measures of liberalization of the aviation market and the impacts of COVID-19 on airline passenger and freight capacity in Australia. ". It stands by my opinion that this is a mistake. b / In my opinion, the literature is still small, especially when looking at the research topic. Opinions of other authors contained in the literature are necessary here, because the level of lberalization is assessed. My other remarks were received with good attention.

Author Response

The authors would like to thank the reviewer for their thorough reviews, including helpful comments and suggestions. Our response to the reviewer comments is given below.

Comment

Response to a reviewer comment

Abstract - my attention has not been taken into account, there is still a sentence with 3 equivalent goals: "This paper examines air transport services liberalization, alternative measures of liberalization of the aviation market and the impacts of COVID-19 on airline passenger and freight capacity in Australia. ". It stands by my opinion that this is a mistake.

The Abstract has been revised.

This paper examines an assessment of the level of air transport services liberalization in Australia in order to generate recommendations on what key market access features of Air Services Agreements should be revised to reflect the changes in air transport characteristics, including the increase in air cargo traffic during the COVID-19 period.  

In my opinion, the literature is still small, especially when looking at the research topic. Opinions of other authors contained in the literature are necessary here, because the level of liberalization is assessed

The paper has been revised and the relevant references have been added.

Please find the information on lines 96-111 and 134-139

Reviewer 3 Report

No remarks.

Author Response

The authors would like to thank the reviewer for their thorough reviews, including helpful comments and suggestions.